# Exploring the Pathological Effect of Aβ42 Oligomers on Neural Networks in Primary Cortical Neuron Culture

**DOI:** 10.3390/ijms24076641

**Published:** 2023-04-02

**Authors:** Dulguun Ganbat, Jae Kyong Jeon, Yunjong Lee, Sang Seong Kim

**Affiliations:** 1Department of Pharmacy, Hanyang University, Ansan 15588, Republic of Korea; 2Department of Pharmacology, School of Medicine, Sungkyunkwan University, Suwon 16419, Republic of Korea

**Keywords:** Alzheimer’s disease, Aβ42, oligomer, high-density multielectrode array, neuronal network, graph theory, center of activity trajectory

## Abstract

Alzheimer’s disease (AD) is a multifactorial disorder that affects cognitive functioning, behavior, and neuronal properties. The neuronal dysfunction is primarily responsible for cognitive decline in AD patients, with many causal factors including plaque accumulation of Aβ42. Neural hyperactivity induced by Aβ42 deposition causes abnormalities in neural networks, leading to alterations in synaptic activity and interneuron dysfunction. Even though neuroimaging techniques elucidated the underlying mechanism of neural connectivity, precise understanding at the cellular level is still elusive. Previous multielectrode array studies have examined the neuronal network modulation in in vitro cultures revealing the relevance of ion channels and the chemical modulators in the presence of Aβ42. In this study, we investigated neuronal connectivity and dynamic changes using a high-density multielectrode array, particularly looking at network-wide parameter changes over time. By comparing the neuronal network between normal and Aβ42treated neuronal cultures, it was possible to discover the direct pathological effect of the Aβ42 oligomer altering the network characteristics. The detrimental effects of the Aβ42 oligomer included not only a decline in spike activation but also a qualitative impairment in neural connectivity as well as a disorientation of dispersibility. As a result, this will improve our understanding of how neural networks are modified during AD progression.

## 1. Introduction

Alzheimer’s disease (AD) has an adverse impact on normal cognitive functioning in memory, reasoning, and even behavior, making it the most debilitating among neurodegenerative diseases [1]. AD is a disorder characterized by a multitude of factors, among which amyloid beta (Aβ) has been demonstrated to contribute significantly to its functional and morphological effects on neuronal properties [2]. In both clinical and experimental settings, paradoxical neural activity is frequently observed at different stages of AD progression [3]. The majority of patients with mild cognitive impairment (MCI) experience epileptic discharges that are followed by cognitive impairment later in life [4,5]. Several studies suggest that pathogenic Aβ42 oligomers, formed by the endosomal proteolytic cleavage of the amyloid precursor protein (APP), cause hyperexcitability in neuronal networks [6,7]. Induced by neural hyperactivity, Aβ42 deposition and tau spread mutually intensify the processes [8]. As part of the pathophysiology of Aβ42, its influence on synaptic modification has been well studied whether it causes presynaptic facilitation or postsynaptic depression [9]. Furthermore, it has been demonstrated that alterations in synaptic activity along with interneuron dysfunction may lead to network aberrations in the presence of APP or Aβ42 [10,11]. Consequently, it is evident that the Aβ42 oligomer can cause abnormalities in neural networks. To gain a more global understanding of cortical connectivity during AD progression, neuroimaging techniques such as diffusion tensor imaging, functional magnetic resonance imaging, and electroencephalography are applied clinically to dementia patients in association with their cognitive status. In contrast to the conventional method, a dissociated neuron culture on a high-density multielectrode array (HD MEA) provides a more controlled means of observing neural network formation in a temporal phase, thus providing insight into neural network alteration [12]. A dissociated neuron culture on an HD MEA can also give insight into the neural network alteration under experimental control by observing network formation from a spatiotemporal perspective. Even though artificial, this neural culture also shares the intrinsic properties of the brain [13]. Having acquired neural signals through an HD MEA, the task of efficiently extracting topological features from Big Data and representing the dynamic status of the network still remains. Amongst several methodologies of connectivity analysis from brain imaging, graph theory is considered a superior metric informing the organizational state of the neural network in terms of integration, segregation, and strength both at structural and functional levels [14,15].

In this study, the dynamic changes in neuronal networks are demonstrated by Aβ42 oligomer exposure over time. Through the analysis of a few critical network parameters using graph theory, the consolidation and dissociation of neural networks are quantitatively assessed. Additionally, a center of activity trajectory (CAT) analysis is conducted to determine the distribution of network burst trajectories and the speed at which neurons transmit signals in culture arenas. It is therefore evident from the HD MEA recordings that the increasing inefficiency of neural networks induced by the Aβ42 oligomer can facilitate a better understanding of how neural networks are affected during AD progression.

## 2. Results

### 2.1. Characteristics of Cortical Neuron Cultures over Culture Ages in Aβ42 Application

It is known that Aβ42 alters neuronal activity depending on its extracellular concentration, causing either presynaptic facilitation or postsynaptic depression [9]. The 10 µM Aβ42 oligomer applied in the present study was proved neurotoxic in the previous study [16]. The mean firing rate (MFR) change was evident in both the visual inspection (Figure 1A) and the violin charts (Figure 1B). In both cultures, the amount of spike activity reflected from the MFR increased steadily as it aged (Figure 1B). In comparison, the MFR of the Aβ42 culture was higher than the control in amount and variation at least until DIV10, which implied a faster initial maturation corresponding to the previous observations of neuronal hyperactivation in the Aβ42 culture [17,18]. It is also noteworthy that the interspike interval (ISI) values, as another indicator of neuronal excitability and connectivity, gradually decreased in both cultures as they aged (Figure 1C). The declining pattern between the Aβ42 and control cultures had a slight difference when the average ISI dropped to 35.68% between DIV1 and 4 in the control compared to 64.02% in the Aβ42 culture. After DIV4, the average ISI remained relatively constant in the control while that of Aβ42 remained at all times higher than the control values. The difference in the ISI decline pattern suggests that the Aβ42 treatment affects synaptic communication at a slower rate and lowers the receptiveness of information exchange among neurons.

### 2.2. An Abnormality in Functional Connectivity in Primary Cortical Neuron Cultures Caused by Aβ42 Oligomers

Figure 2A illustrates distinct dissimilarities between the control and Aβ42 cultures based on the graphical perspective of the connectivity map. It is interesting to note that compared to the quantitative increase in nodal links connecting neurons in control cultures over culture ages, Aβ42 connectivity showed a biphasic pattern with an initial increase in complexity until DIV10 and then a decline until DIV16. This is similar to the neural burst and MFR observed in Figure 1. CC represents the ability of neurons to form local clusters. In this respect, the Aβ42 culture retained robust clustering throughout culture days with higher CC values than the control (Figure 2B). Interestingly, CC values in the control were consistent from DIV4 to 10 then abruptly increased from DIV13. However, PL values indicated a more dynamic change. From DIV4 to 13, the PL values in the control decreased gradually (Figure 2C). Contrary to this, the PL of the Aβ42 culture reflected the connectivity map in a biphasic pattern with a steep incline up to DIV4 and then a gradual decline afterward. Because PL refers to the minimum number of steps necessary to reach one node from another, a shorter PL signifies more efficient connections within the network, or in other words better small-worldness [15]. In this case, the control culture developed network efficiency with age. The lowest PL occurring at the same time as the highest CC in DIV13 of the control culture implied that dissociated neurons reached a state of self-organized criticality, in which a system can respond rapidly to changes in the environment while maintaining stability and robustness despite any external or internal perturbations [19]. However, the gradual increase in PL values from DIV4 to 10 in the Aβ42 culture indicated the escalating inefficiency in network formation. The later decline in DIV13 to 16 possibly resulted from the decrease in the overall network size itself noticeable in the connectivity map as well as in ND values (Figure 2A,D). As a result, the Aβ42 culture did not achieve a critical state unlike the control. Another network parameter, ND also provides a valuable tool to understand network organization (Figure 2D). In the Aβ42 culture, due to an almost six-fold increase in the average ND from DIV4 to 7, pruning of connecting branches within nodes was not executed optimally, resulting in an extra burden for neurons. NDs in the control culture, however, showed a gradual increase in conjunction with a decline in PL values. According to all network parameter changes in the control culture, the network was maturing as nonessential connections were pruned, local connections were increased, and shortcuts were developed with long distance nodes. As illustrated in Appendix A, the histogram plot of the PL distribution demonstrated the maturation of neural networks in a more sophisticated manner. The Aβ42 culture demonstrated earlier neural linkage in all ranges of PL distribution compared with control cultures that initiated this later. The median PL histogram for the control culture remained at 2.75 links during DIV13 when the network reached the critical stage. In the Aβ42 culture, however, the median PL histogram increased from 2.25 (DIV7) to 3.25 (DIV10) and 2.75 (DIV13). The shift in the median in the Aβ42 culture resulted in fatter tails in the PL distribution histogram than in the control, meaning the shortcut links to a distant node are longer. A noticeable growth of links in the control from previous DIVs was apparent in DIV16, which kept the median value at the same level, showing that the network’s integrity was stable. The Aβ42 culture, on the other hand, did not display a change in the tail pattern of PL distribution, indicating that network size expansion was hampered. The fat tail in higher ND links was also observed in the Aβ42 culture (Appendix A). Based on all of the topological parameters, it is clear that the Aβ42 oligomer impairs both qualitative and quantitative neural network maturation without establishing network criticality.

### 2.3. Modification of Spike Burst Patterns Affected by Aβ42 Application

Figure 3 illustrates spike bursts in cultures that occur when neurons fire rapid action potentials within a narrow timeframe. As cultures became older, spike bursts and network bursts were observed (Figure 3A). In control cultures, bursts and frequencies followed a continuous upward trend except for DIV13 (Figure 3B,C). In contrast, spike burst duration decreased from DIV4 to DIV16 except for DIV13 (Figure 3D). A similar pattern was observed for spike network bursts as well, in that their number and frequency gradually increased with a decline in duration time except for DIV13 (Figure 3E–G). The burst pattern of the Aβ42 culture, however, did not show distinct characteristics. The number of spike bursts decreased from DIV1 to 4, but there was no difference between DIV4 and 7 (Figure 3B). The increase from DIV7 to 10 was reversed when it reached DIV13. Spike durations tended to decrease steadily except for DIV1 (Figure 3D). When DIV1 was ignored, a biphasic pattern in the number and frequency of Aβ42 culture network bursts was observed where the initial incline reversed to decline at DIV10 (Figure 3E,F). In contrast, when we excluded DIV1, the network duration decreased continuously, similarly to spike burst patterns (Figure 3G).

### 2.4. Network-Wide Properties of Spatiotemporal Signal Transduction in the Dissociated Culture Arena Using CAT Analysis

In the context of neuronal networks, where activity patterns are the result of complex interactions between the neurons involved, the dynamics of when and where these neurons fire, or in other words the spatiotemporal modality, needs to be considered comprehensively. By including the vector summation of neural activities in the center of activity (CA), the flow of population activity over a short period can be analyzed [20]. A CA will be located closer to the center of the arena if the neural activity is more homogeneous, as the individual vector is calculated from the center of the arena. Thus, sequential tracing of CA can provide valuable information regarding the spatiotemporal trajectory of neural bursts, such as coherence, speed, and dispersibility [21]. In the control, there was a gradual increase in the number of network bursts from 0 to over 50 counts as the culture aged, whereas that of the Aβ42 culture was relatively constant at between 20 and 30 counts (Figure 3E). The total density of CATs was also affected by network-burst count differences (Figure 4A). In the control, the CA position at the end of the trajectory (yellow) shifted towards the center of the arena from DIV7 onwards. A vector space can be used to visualize the travel route from beginning to end for each CAT (Appendix A). In the Aβ42 culture, however, the CATs displayed disorientation rather than converging on the central spot in the culture arena. Based on this result, the neural firing in the control is homogeneous even at relatively early culture stages at DIV7, while the firing in the Aβ42 culture remains inhomogeneous throughout the entire culture period. The Aβ42 culture results in shorter CAT duration and faster velocity possibly because the neural activity is localized off the center pattern of CAT (Figure 4B,C). CC values are higher in this condition as neural signals are limited to a local compartment, thereby explaining the difficulty in reaching distant neurons (Figure 2B).

## 3. Discussion

It is inherent for neurons to construct a neuronal network that is optimized for neuronal communications and resilience to minimize perturbations. In the brains of AD patients, however, it is noteworthy to find an irregular activation of neurons and disruption of the neural network organization [22]. Previous studies using an MEA have characterized neural network dysfunction by Aβ42 oligomer treatment [23,24]. Hamid et al. observed a 60% reduction in spike rate when 5 µM of Aβ42 oligomer was applied to a neuronal culture on a 60-electrode MEA [23]. Another HD MEA study demonstrated an approximately 50% reduction in the MFR in DIV24 hippocampal neuronal cultures after 26 h of treatment with 0.1 µM of Aβ42 oligomer [24]. At 1 µM it decreased by almost 23.5%, and at 10 µM, spike activity was eliminated. While applying 10 µM of Aβ42 oligomer to neuronal cultures from DIV1 to 16, we still observed spikes, possibly because cultures were taken at a relatively young age from DIV1 to 16. The vitality of younger neurons as well as various cell types in the cortical culture could potentially prolong neuron survival under the toxicity of the Aβ42 oligomer. In this study, we further elaborated on neural connectivity and dynamic changes, particularly around network-wide parameter changes. By comparing the neuronal network with the same culture condition, it was possible to see the direct pathological effect of the Aβ42 oligomer. Based on CC, PL, and ND value changes, the topological and qualitative properties were analyzed for the communication efficiency and optimization ability of the neural network. In normal neuron culture, it took 13 days to build up network optimization, or, in other words, criticality. Although the quantitative network size of neurons cultured with the Aβ42 oligomer was much larger than that of the control cultures, their network became directed toward anomalies instead. Normally, the burst frequency increases while the burst duration decreases as the cultures mature, indicating a transition from a low frequency, high duration burst pattern to a high frequency, low duration burst pattern. In this study, we observed the same pattern in the normal neuronal culture. On the contrary, no noticeable tendency of burst parameters could be found in the Aβ42-treated culture implying the imbalance of excitatory and inhibitory synapses within the network, or changes in the intrinsic properties of the neurons. The CAT analysis also revealed a novel finding about the intrinsic nature of network structure. It had been challenging to assess the network-wide synchrony of neuronal bursts in a singular parameter containing spatiotemporal characteristics. Through CAT analysis, more homogeneous neuronal bursts were observed in the normal neurons at a relatively early stage of development. The eccentric localization of the CA in the Aβ42 culture indicates the fractured nature of the neural network in AD, which accounts for the scattered activation areas in other brain imaging studies.

By measuring neuronal connectivity meticulously with an HD MEA, we were able to identify how neuronal connectivity developed over time. A graph theory and CAT analysis approach was employed to quantify the efficiency and synchrony of neural networks as representative parameters for defining critical states. The efficacy of therapeutic candidates can be evaluated in future studies by observing and comparing other AD factors such as tau or relevant genes on neuronal network alterations in a similar recording setting. This will allow us to screen more effective drug candidates and determine a more accurate therapeutic window to treat AD.

## 4. Materials and Methods

### 4.1. Aβ42 Oligomer Preparation

The peptide corresponding to human Aβ42 oligomer (Anaspec, AS-64129-1, 1 mg, Fremont, CA, USA) was dissolved in 100 μL of DMSO by vortexing for 30 min at room temperature, and then the solution was added to 900 μL of PBS for incubation at 4 °C for 24 h.

### 4.2. Primary Neuron Culture

Dissection medium Neurobasal Media (NBM) consisted of 45 mL Neurobasal Medium A, 1 mL B27 (50×), 0.5 mM Glutamine sol, 25 μM Glutamate, 5 mL Horse serum, and 500 μL penicillin/streptomycin. Culture medium consisted of 50 mL Neurobasal Medium A, 1 mL B27, 0.5 mM Glutamine sol, 500 μL penicillin/streptomycin, and 50 μL HEPES. The Biochip chamber (3Brain, Arena, Zurich, Switzerland) was cleaned, filled with 70% ethanol for 20–30 min, rinsed 3–4 times with autoclaved DDW, and dried in a clean bench overnight with NBM. The following day, 30–90 μL filtered PDLO which was dissolved in borate buffer on the active surface of the Biochip was added and placed overnight in the incubator. The Biochip was washed with autoclaved DDW 3 times before cell seeding. Primary cortical and hippocampal neuron cultures were prepared from postnatal 0-day mouse pups. Pups were decapitated with sterilized scissors and the whole brain was removed. The removed brain was chilled in a cold neurobasal medium with papain 0.003 g/mL solution at 4 °C in a 35 mm diameter dish. Surrounding meninges and excess white matter were pulled out under the microscope (Inverted microscope, Nikon, Japan) in the same medium as a second dish at 4 °C. The cortex and hippocampus parts were isolated from other parts of the brain, washed with NBM and papain solution, and minced into small pieces. The minced tissues were transferred into a 15 mL tube and incubated for 30 min in a 37 °C water bath. After this, the tube was inverted gently every 5 min to be mixed. The tissues were washed with HBSS twice. After settling, the cortex and hippocampi tissues were transferred into prewarmed NBM and triturated 20–30 times using a fire-polished Pasteur pipette. The number of cells was counted and 30–90 µL drops of the cells were plated in the Biochip, which contains ~1000–1500 cells/µL (incubated at 37 °C in 5% CO_2_). To compare neuronal network activity between the control and Aβ42 treatment cultures over time, two sets of primary cortical neurons were prepared on 64 × 64 HD MEA chips with the same number of neurons, culture conditions, and measurement periods until DIV16. The whole medium was replaced with a fresh feeding medium every 3 days.

### 4.3. Neuronal Spike Recording with HD MEA and Data Analysis

HD MEA recording with 4096 electrodes in CMOS Biochip (BiocamX; 3Brain GmbH, Zurich, Switzerland) was conducted at a sampling rate of 10 KHz. The active electrode which is 21 μm × 21 μm in size and 42 μm in pitch was implanted in the array with a 64 × 64 grid (2.67 × 2.67 mm^2^) centered on a working area (6 × 6 mm^2^). The spontaneous neuronal spikes for each culture were recorded for 5 min at the same time of culture days. All recordings were conducted and analyzed by Brainwave software (3Brain GmbH, Zurich, Switzerland). The CAT analysis in Brainwave adopted the algorithm of Gandolfo et al. [21].

## 5. Conclusions

The study demonstrates the irregular activation of neurons and the disruption in neural network organization by Aβ42 oligomer treatment using an HD MEA. An Aβ42 neuronal culture displays significant impairments in basic topological properties over time, including spike frequency and duration. Network analysis based on graph theory also indicates a disruption in the neural network as a result of changes in network parameters, such as CC, PL, and ND. Furthermore, CAT analysis reveals asynchronous communication between neurons and inhomogeneous network bursts. Through this study, we will gain a better understanding of how neural networks change as AD progresses.

## Figures and Tables

**Figure 1 ijms-24-06641-f001:**
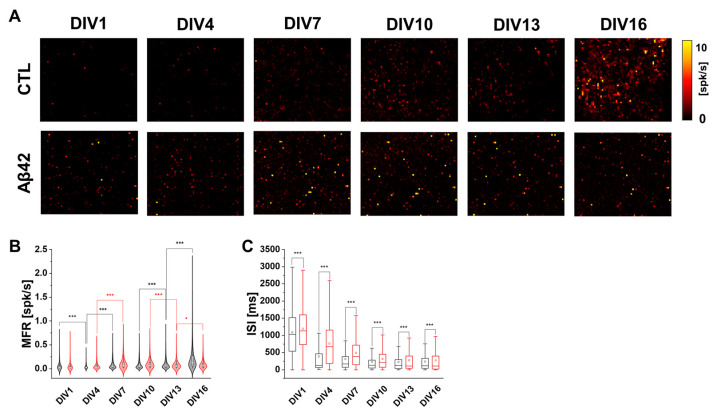
Spike activation property of developing neuronal cultures in HD MEA recording measurements. (**A**) MFR Activity map of spontaneous neuronal activation during 5 min recordings. Control (CTL) and Aβ42 oligomer treatment groups in DIV1, 4, 7, 10, 13, and 16. The intensity scale ranges from 0 to 10 spikes per second. (**B**) Violin plot of mean firing rate (MFR) for 5 min recordings for control (black) and Aβ42 (red) cultures during DIVs. The white dot represents the median and the thick white bar in the center represents the interquartile range. The thin black line represents the 1.5× interquartile range. (**C**) Box plot of interspike intervals (ISI) in milliseconds for 5 min recordings for control (black) and Aβ42 (red) cultures during DIVs. The lower quartile as the borderline of the box nearest to zero expresses the 25th percentile, whereas the upper quartile as the borderline of the box farthest from zero indicates the 75th percentile. Error bars show SEM. *** *p* < 0.005, * *p* < 0.05; unpaired, two-tailed *t*-test with Welch’s correction.

**Figure 2 ijms-24-06641-f002:**
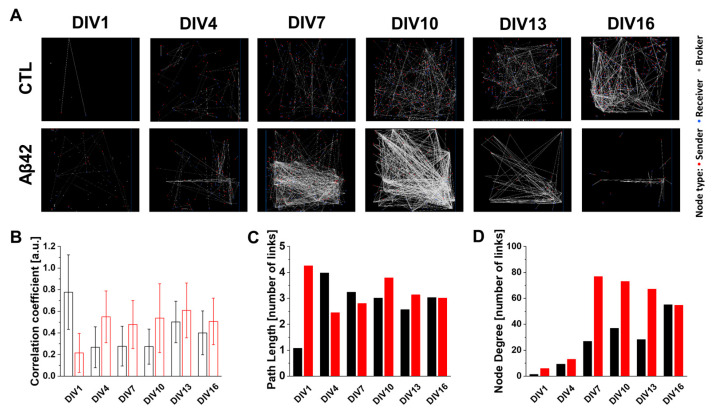
Neuronal connectivity map and its characterization based on graph theory. (**A**) Neuronal connectivity map in spontaneous neuronal activation during 5 min recordings. Control (CTL) and Aβ42 oligomer treatment groups in DIV1, 4, 7, 10, 13, and 16. The red dot represents a node of the sender, blue for the receiver, and gray for the broker. The white line describes the connection between nodes. (**B**) Average of clustering coefficients from neuronal connections for 5 min recordings for control (black) and Aβ42 (red) cultures during DIVs. The line represents a standard deviation. (**C**) Histogram plot of average path length in a number of links. (**D**) Histogram plot of node degrees in a number of links.

**Figure 3 ijms-24-06641-f003:**
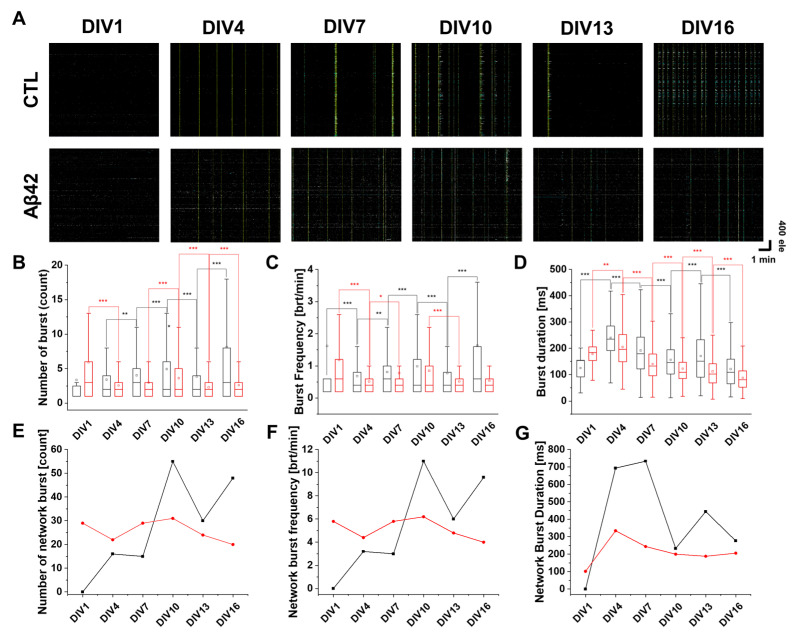
Spike burst analysis in the developing neuronal cultures. (**A**) Raster plot of neuronal spiking from 4096 electrodes (y-axis) during 5 min recordings (x-axis). Control (CTL) and Aβ42 oligomer treatment groups in DIV1, 4, 7, 10, 13, and 16. The scale represents one minute and 400 electrodes. The mustard color represents spike network bursts, and blue shows spike bursts. (**B**) Box plot of a number of spike bursts for control (black) and Aβ42 oligomer treatment (red) groups. (**C**) Box plot of spike burst frequency showing number of bursts per minute. (**D**) Box plot of spike burst duration in milliseconds. (**E**) Line plot of the number of network bursts for control (black) and Aβ42 oligomer treatment (red) groups. (**F**) Line plot of network burst frequency in the number of network bursts per minute. (**G**) Line plot of network burst duration in milliseconds. In all the box plots above, the lower quartile as the borderline of the box nearest to zero expresses the 25th percentile, whereas the upper quartile as the borderline of the box farthest from zero indicates the 75th percentile. Error bars show SEM. *** *p* < 0.005, ** *p* < 0.01, * *p* < 0.05; unpaired, two-tailed *t*-test with Welch’s correction.

**Figure 4 ijms-24-06641-f004:**
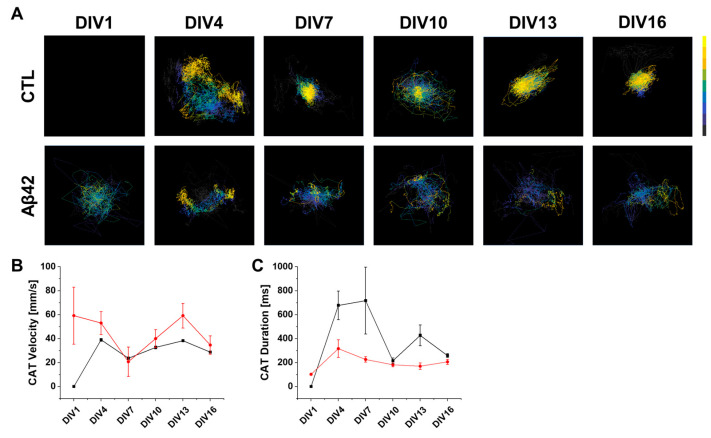
CAT analysis in the developing neuronal cultures. (**A**) CAT analysis images of control (CTL) and Aβ42 oligomer treatment groups in DIV1, 4, 7, 10, 13, and 16. The color scale bar represents initiation (gray) to termination (yellow). (**B**) Line plot of CAT velocity with variation (vertical line) in millimeters per second for control (black) and Aβ42 oligomer treatment (red) groups. (**C**) Line plot of CAT duration with variation (vertical line) in milliseconds.

## Data Availability

All the data can be shared through the database of MDPI.

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
