# Peer review of "Exploring the Pathological Effect of Aβ42 Oligomers on Neural Networks in Primary Cortical Neuron Culture"

_ijms, 2023, doi:10.3390/ijms24076641_

Round 1

Reviewer 1 Report

Reviewer’s Comments:

The manuscript “Functional aberration of cortical neuronal network induced by Aβ42 oligomer” is a very interesting work. In this work, Alzheimer's disease (AD) is a multifactorial disorder that affects cognitive functioning, behavior, and neuronal properties. The neuronal dysfunction is primarily responsible for cognitive decline in AD patients, with many causal factors including plaque accumulation of Aβ42. Neural hyperactivity induced by Aβ42 deposition cause abnormalities in neural networks, leading to alterations in synaptic activity and interneuron dysfunction. Even though neuroimaging techniques elucidated the underlying mechanism in the neural connectivity, precise understanding in cellular level is still elusive. Previously, a few multielectrode array studies examined the neuronal network modulation in vitro cultures revealing relevance of ion channels and the chemical modulators in the presence of Aβ42. In this study, we investigated neuronal connectivity and dynamic changes with high density multielectrode array, particularly in relation to network-wide parameter changes over time. While I believe this topic is of great interest to our readers, I think it needs major revision before it is ready for publication. So, I recommend this manuscript for publication with major revisions.

1. In this manuscript, the authors did not explain the importance of the oligomer in the introduction part. The authors should explain the importance of oligomers.

2) Title: The title of the manuscript is not impressive. It should be modified or rewritten it.

3) Correct the following statement “The application of graph theory and center of activity trajectory analysis assessed the consolidation and disassociation of neural networks under Aβ42 oligomer exposure over time. This result can enhance our understanding of how neural networks are affected during AD progression”.

4) Keywords: The oligomer is missing in the keywords. So, modify the keywords.

5) Introduction part is not impressive. The references cited are very old. So, Improve it with some latest literature like 10.1016/j.molstruc.2021.131136, 10.1002/slct.202101634

6) The authors should explain the following statement with recent references, “An aging brain or a neurotoxic state, which is the hallmark of Alzheimer's disease, deteriorates network stability”.

7) Add space between magnitude and unit. For example, in synthesis “21.96g” should be 21.96 g. Make the corrections throughout the manuscript regarding values and units.

8) The author should provide reason about this statement “Through CAT analysis, more homogeneous neuronal bursts were observed in the normal neurons at a relatively early stage of development”.

9. Comparison of the present results with other similar findings in the literature should be discussed in more detail. This is necessary in order to place this work together with other work in the field and to give more credibility to the present results.

10) Conclusion part is very long. Make it brief and improve by adding the results of your studies.

11) There are many grammatic mistakes. Improve the English grammar of the manuscript.

Author Response

  1. In this manuscript, the authors did not explain the importance of the oligomer in the introduction part. The authors should explain the importance of oligomers.

The importance of the oligomer has been addressed in the introduction like below.

AD is a multifactorial disorder characterized by a multitude of factors, among which amyloid beta (Aβ) has been demonstrated to contribute significantly to its functional and morphological effects on neuronal properties [2].

Several studies suggest that pathogenic Aβ42 oligomers, formed by the endosomal proteolytic cleavage of the amyloid precursor protein (APP), cause hyperexcitability in neuronal networks [7]. Induced by neural hyperactivity, Aβ42 deposition and tau spread are mutually intensifying the processes [8]. As part of the pathophysiology of Aβ42, its influence on synaptic modification has been well studied whether it causes presynaptic facilitation or postsynaptic depression [9]. Furthermore, it has been demonstrated that alterations in synaptic activity along with interneuron dysfunction may lead to network aberrations in the presence of APP or Aβ42 [10, 11]. Consequently, it is evident that Aβ42 oligomer can cause abnormalities in neural networks.

If an additional explanation is necessary, please let me know. In that case, it would be worrisome to make the introduction too lengthy.

2) Title: The title of the manuscript is not impressive. It should be modified or rewritten it.

It’s rewritten as below.

Exploring the Pathological Effect of Aβ42 Oligomers on Neural Networks in Primary Cortical Neuron Culture

3) Correct the following statement “The application of graph theory and center of activity trajectory analysis assessed the consolidation and disassociation of neural networks under Aβ42 oligomer exposure over time. This result can enhance our understanding of how neural networks are affected during AD progression”.

It’s corrected as below.

As a result, this will improve our understanding of how neural networks are during AD progression.

4) Keywords: The oligomer is missing in the keywords. So, modify the keywords.

Added.

5) Introduction part is not impressive. The references cited are very old. So, Improve it with some latest literature like 10.1016/j.molstruc.2021.131136, 10.1002/slct.202101634

The suggested papers are about the new series of bis-thioureas synthesized as acetylcholinesterase enzyme inhibition activity. Since the scope of current manuscript does not cover a specific drug discovery of AD, the recommended papers seems not relevant to be incorporated in the introduction. Instead, we updated the references to more recent ones at least after 2000 up to 2021.

6) The authors should explain the following statement with recent references, “An aging brain or a neurotoxic state, which is the hallmark of Alzheimer's disease, deteriorates network stability”.

A recent reference is added.

7) Add space between magnitude and unit. For example, in synthesis “21.96g” should be 21.96 g. Make the corrections throughout the manuscript regarding values and units.

All the units are checked and corrected.

8) The author should provide reason about this statement “Through CAT analysis, more homogeneous neuronal bursts were observed in the normal neurons at a relatively early stage of development”.

Since the reason underlying the statement “Through CAT analysis, more homogeneous neuronal bursts were observed in the normal neurons at a relatively early stage of development” has been explained in the result, it seems to be redundant to reiterate the explanation in discussion again. Even so, please let us know if it looks still insufficient.

In result section,

The total density of CATs was also affected by network burst count differences (Figure 4A). In control, the CA position at the end of trajectory (yellow) shifted towards the center of the arena from DIV 7 forward. A vector space can be used to visualize the travel route from beginning to end for each individual CAT (Supplementary figure 3). With the disoriented end point outside the center in Aβ42 culture, the dispersion of CAT was more spread out network-wide. Based on this result, the neural firing in control is homogeneous even at relatively early culture stages at DIV7, while the firing in Aβ42 culture remains inhomogeneous throughout the entire culture period.

  1. Comparison of the present results with other similar findings in the literature should be discussed in more detail. This is necessary in order to place this work together with other work in the field and to give more credibility to the present results.

Comparison of the present results with other similar findings are described in the updated discussion as below.

In the brains of AD patients, however, it is noteworthy to find irregular activation of neu-rons and disruption of neural network organization [22]. Previously, a few studies using MEA have characterized neural network dysfunction by Aβ42 oligomer treatment [23, 24]. Hamid et al. observed a 60 % reduction in spike rate when 5 µM Aβ42 oligomer was ap-plied in neuronal culture on 60 electrodes MEA [23]. Another HD MEA study demon-strated an approximately 50 % reduction in MFR in DIV24 hippocampal neuronal cul-tures after 26 hours of treatment with 0.1 µM Aβ42 oligomer [24]. Upon 1 µM, it decreased by almost 23.5 %, and at 10 µM, spike activity was eliminated. While applying 10 µM Aβ42 oligomer to neuronal cultures from DIV1 to 16, we still observed spikes, possibly because cultures were taken at a relatively young age from DIV1 to 16. The vitality of younger neurons as well as various cell types in the cortical culture could potentially pro-long neurons' survival under the toxicity of Aβ42 oligomer. In this study, we further elab-orated on neural connectivity and dynamic changes, particularly about network-wide parameter changes.

10) Conclusion part is very long. Make it brief and improve by adding the results of your studies.

Unnecessary sentences are deleted and replaced with the discussion of results. By this way overall length of discussion part became shorter.

11) There are many grammatic mistakes. Improve the English grammar of the manuscript.

All grammatical error are corrected.

Reviewer 2 Report

This paper discussed the functional aberrations of cortical neuronal networks induced by the Aβ42 oligomer Aβ42. Overall, the paper is interesting. The authors are advised to consider the following suggestions to further improve the paper’s quality.

(1) It is recommended to avoid lumped references in the Introduction section, such as [15-17].

(2) The quality of the figures needs to be further improved. Among them, the display of many details of the diagram was not clear. For example, the words in Figure 3B were not clear. The authors are advised to enlarge all the figures.

(3) The authors are suggested to add a Summary/Conclusions section, which should include the contribution of this paper and the major findings.

(4) The whole paper needs to further polish. The current version still has grammar mistakes and typo errors.

Author Response

  • It is recommended to avoid lumped references in the Introduction section, such as [15-17].

The number of references is trimmed to 24 from 27 including 16.

  • The quality of the figures needs to be further improved. Among them, the display of many details of the diagram was not clear. For example, the words in Figure 3B were not clear. The authors are advised to enlarge all the figures.

All figures are modified for a better visual recognition.

  • The authors are suggested to add a Summary/Conclusions section, which should include the contribution of this paper and the major findings.

Conclusions are inserted as below.

The study demonstrates the irregular activation of neurons and the disruption of neural network organization by Aβ42 oligomer treatment using HD MEA. Aβ42 neuronal culture displays significant impairments in basic topological properties over time, including spike frequency and duration. Network analysis based on graph theory also indicates a disruption in the neural network as a result of changes in network parameters, such as CC, PL, and ND. Furthermore, CAT analysis reveals asynchronous communication be-tween neurons and inhomogeneous network bursts. Through this study, we will gain a better understanding of how neural networks change as AD progresses.

  • The whole paper needs to further polish. The current version still has grammar mistakes and typo errors.

All grammatical and typo errors are corrected.